# Laboratory Diagnostics of *Rickettsia* Infections in Denmark 2008–2015

**DOI:** 10.3390/biology9060133

**Published:** 2020-06-19

**Authors:** Susanne Schjørring, Martin Tugwell Jepsen, Camilla Adler Sørensen, Palle Valentiner-Branth, Bjørn Kantsø, Randi Føns Petersen, Ole Skovgaard, Karen A. Krogfelt

**Affiliations:** 1Department of Bacteria, Parasites and Fungi, Statens Serum Institut (SSI), 2300 Copenhagen, Denmark; ssc@ssi.dk (S.S.); martintjepsen@gmail.com (M.T.J.); rfp@ssi.dk (R.F.P.); 2European Program for Public Health Microbiology Training (EUPHEM), European Centre for Disease Prevention and Control (ECDC), 27180 Solnar, Sweden; 3Scandtick Innovation, Project Group, InterReg, 551 11 Jönköping, Sweden; cas2300@gmail.com; 4Virus and Microbiological Special Diagnostics, Statens Serum Institut (SSI), 2300 Copenhagen, Denmark; bjoern@kantsoe.net; 5Department of Infectious Disease Epidemiology and Prevention, Statens Serum Institut (SSI), 2300 Copenhagen, Denmark; pvb@ssi.dk; 6Department of Science and Environment, Roskilde University, 4000 Roskilde, Denmark

**Keywords:** rickettsiosis, *Rickettsia* spp., diagnostics of rickettsiosis, serology, PCR, tick-borne infections, vector-borne infections

## Abstract

Rickettsiosis is a vector-borne disease caused by bacterial species in the genus *Rickettsia*. Ticks in Scandinavia are reported to be infected with *Rickettsia*, yet only a few Scandinavian human cases are described, and rickettsiosis is poorly understood. The aim of this study was to determine the prevalence of rickettsiosis in Denmark based on laboratory findings. We found that in the Danish individuals who tested positive for *Rickettsia* by serology, the majority (86%; 484/561) of the infections belonged to the spotted fever group. In contrast, we could confirm 13 of 41 (32%) PCR-positive individuals by sequencing and identified all of these as *R. africae*, indicating infections after travel exposure. These 13 samples were collected from wound/skin material. In Denmark, approximately 85 individuals test positive for *Rickettsia* spp. annually, giving an estimated 26% (561/2147) annual prevalence among those suspected of rickettsiosis after tick bites. However, without clinical data and a history of travel exposure, a true estimation of rickettsiosis acquired endemically by tick bites cannot be made. Therefore, we recommend that both clinical data and specific travel exposure be included in a surveillance system of *Rickettsia* infections.

## 1. Introduction

Tick-borne rickettsiosis is caused by obligate intracellular bacteria belonging to the genus *Rickettsia* and can be classified into two major disease groups: the spotted fever group (SFG), and the typhus group (TG). *Rickettsia* spp. are among the oldest known agents for vector-borne diseases, yet recently, an increase in incidence has been observed in the USA [1]. Variations in annual incidence may be attributed to a range of factors, such as human interaction with tick habitats, vector and host dynamics, climatic or ecological changes, increased awareness and testing of tick-borne diseases, or changes in surveillance practices [1].

*Rickettsia* spp. belonging to the SFG are often attributed to causing disease worldwide. The pathophysiology is characterized by invasion and replication in vascular endothelial cells, causing varying degrees of vasculitis in small to medium-sized blood vessels and resulting in symptoms such as fever, rash, headache, myalgia, arthralgia, and sometimes necrotizing eschar (“tache noir”). Clinical severity is often associated with the underlying species and ranges from potentially fatal diseases such as the Rocky Mountain spotted fever caused by *Rickettsia rickettsii* to the more benign African tick-bite fever caused by *Rickettsia africae* [1,2,3].

Until recently, the diagnosis of tick-borne SFG rickettsiosis was confirmed almost exclusively by serological methods because culturing needs specialized facilities. The oldest method for testing, the Weil–Felix test, is still used in developing countries [4,5]. This test is based on the detection of antibodies to formalin-inactivated whole cells of *Proteus* spp. (OX19, OX2, and OXK) that cross-react with *Rickettsia* of the SFG. However, this assay lacks sensitivity and specificity [6]. The microimmunofluorescence (MIF) assay is the current reference method for the detection of antibodies to *Rickettsia* spp., yet antigenic cross-reactions are seen within the SFG. Confirmation of tick-borne rickettsiosis in human samples has been supported and facilitated using molecular methods for identifying *Rickettsia* spp. A consensus on recommendations for the diagnosis of rickettsiosis was presented in “European guidelines for diagnosis of tick-borne diseases”, which supports clinical diagnosis by molecular methods and serology [2].

In Denmark, only a limited number of studies have evaluated *Rickettsia* infections in humans after a tick bite. Furthermore, patients are rarely tested for *Rickettsia* after a tick bite, even though the presence of *Rickettsia* spp. in Denmark has been confirmed in ticks collected from domestic dogs or by flagging [7,8]. *Rickettsia helvetica* was found to be one of the most common pathogens in the *Ixodes ricinus* tick [7], but *Anaplasma phagocytophilum*, *Borrelia burgdorferi*, and *R. helvetica* have also been detected in *I. ricinus* ticks by PCR [9]. Previous studies have shown that Danish roe deer were seropositive for *Borrelia* spp., *A. phagocytophilium*, and tick-borne encephalitis virus (TBEV) [10]. Although there are several reports of ticks colonized by *Rickettsia* spp. [7,8,9] and of *Rickettsia* seropositivity in selected groups of humans [11,12,13,14,15], no systematic analysis of *Rickettsia* spp. prevalence and transmission has been conducted nationwide.

In one study, patients positive for *B. burgdorferi* were screened for *Rickettsia* antibodies, and 12.5% (21/168) were found positive for *R. helvetica* antibodies [11]. Despite a high frequency of tick bites, antibodies against *Rickettsia* were not detected in Danish elite orienteers [12]. *Rickettsia* has also been ruled out for involvement in the pathogenesis of sarcoidosis [13].

Rickettsiosis is potentially underestimated in Danish travelers returning from Africa, Southeastern Europe, and the US, where *Rickettsia africae* is the agent of African tick-bite fever (ATBF) and *R. rickettsii* and *R. conori* are agents of spotted fever group, respectively [6,14,15,16,17]. As rickettsiosis is not notifiable in Denmark, clinical and travel information is often lacking, which may cause trouble when trying to describe the disease prevalence. Furthermore, a recent study on the clinical assessment of rickettsiosis acquired endemically in Denmark suggests that the disease presents with mild symptoms compared with imported *Rickettsia* infections [3]. The quality and precision of national surveillance data depend on clinical and reporting practices for rickettsiosis.

In this study, we assessed samples submitted for routine diagnosis of rickettsiosis in Denmark in the period 2008–2015. Our aim is to enhance the awareness of rickettsiosis after tick bites and provide a basis for recommendations in managing tick-borne infections in Denmark.

## 2. Materials and Methods

### 2.1. Data Collection and Study Population

Data from all samples tested for *Rickettsia* spp. DNA and specific *Rickettsia* antibodies (IgG and IgM) from 1 July 2008 to 19 October 2015 for all of Denmark were retrieved from the laboratory database at Statens Serum Institut (SSI). Age, sex, sample date, as well as travel and clinical information (when provided) were recorded. Clinical and travel data were analyzed and presented by Ocias et al. [3]. The data analysis was performed on samples from individuals with a permanent address in Denmark, excluding Greenland. Statistics and conclusions summarized in this report represent descriptions of the cases captured in the reporting period of this study. In total, 2819 samples obtained from 2240 individuals during this period were analyzed (Figure 1).

### 2.2. Rickettsia Antibody Detection

Detection of IgG and IgM for SFG was performed using Focus Diagnostics’ Immunofluorescent assay (IFA) (IF0100G/IF0100M) with antigens from the spotted fever group (*Rickettsia rickettsii*) and the typhus group (*Rickettsia typhi*). Positive cut-off values were IgG ≥ 1:512 and/or IgM ≥ 1:64 and adjusted to the Danish population and exposure [18].

Additionally, detection of IgG and IgM antibodies against *Bartonella henselae* and *Bartonella quintana* and IgG against *Anaplasma phagocytophilum* was performed as described by the manufacturer (Focus Diagnostics IFA IF1300G/IF1300M and IF1450G respectively). Detection of *Francisella tularensis* was performed by an in-house agglutination test with cut-off ≥ 1:10. The detection of IgG and IgM antibodies against *B. burgdorferi* was performed by an in-house ELISA modified from Hansen et al. [19].

### 2.3. Detection of Rickettsia spp. DNA by PCR Assay and Sequencing

Real-time PCR was used to amplify a 74-bp fragment of the citrate synthase (CS) gene (gltA) present in all *Rickettsia* spp., according to Stenos et al. [20]. DNA from plasma/serum samples were purified by QIAamp DNA Mini Kit (Qiagen 51,306) according to the manufacturer’s instructions. Samples positive for gltA amplification were further analyzed by conventional 16S RNA sequencing using PCR primers (Rick16SF2 5′-ACG CTA TCG GTA TGC TTA ACA CAT G-3′ and Rick16SR2 5′-CAA CTT ACT AAA CCG CCT ACG CAC T-3′) targeting the 16S rRNA gene for species identification. The sequences were compared to the Nucleotide collection (nr/nt) using NCBI BLAST. Positive control DNA was kindly provided by Professor John Stenos [20].

## 3. Results

### 3.1. Diagnostic Samples

In the years 2008–2015, a total of 2819 samples were analyzed with either PCR and/or serology testing (Table 1, Figure 1). The majority of samples were tested for the presence of *Rickettsia* antibodies (2404, 85%), and of these, 1156 (48%) were tested for the presence of additional antibodies against four other tick-borne or related bacteria, namely, *Bartonella*, *Anaplasma*, *Francisella*, and *Borrelia* spp.

Real-time PCR analysis was applied to 415 (15%) samples. These samples were categorized: blood (serum/plasma), unspecified organ biopsy, wound/skin, and unspecified (Figure 1). Most of the samples tested by PCR were serum/plasma (72%; 298/415). Within the category of serum/plasma, 79% (236/298) were obtained in ethylenediaminetetraacetic acid (EDTA) tubes, where only three samples (1%; 3/235) tested positive by PCR. The positive rates for organ biopsy and wound/skin were 13% (2/15) and 46% (37/81), respectively (Figure 1).

16S RNA sequencing was attempted on all samples that tested positive for *Rickettsia* gltA by real-time PCR (n = 43). Thirteen PCR-positive samples were sequenced successfully; the remaining 30 samples did not yield sequence templates of suitable quality. Twelve samples perfectly matched *R. africae*, acc. nos. CP001612 and L36098; the last sample had one mismatch with these sequences, but *R. africae* was still the closest match. Only 5 of these 13 *R. africae*-positive patients had reported travel to Africa, while the rest did not provide travel information. The samples that could be sequenced successfully were all taken from wounds or skin biopsies. The obtained sequences are given in Appendix A.

In total, 769 samples tested positive for *Rickettsia* in the period 2008–2015 in Denmark; of these, 726 (94%) were positive by serology and 43 (6%) by PCR (Figure 1).

### 3.2. Samples Tested for Rickettsia Antibodies

We found a similar distribution between samples positive for *Rickettsia* antibodies that were tested for *Rickettsia* antibodies only (34%, 427/1248) and samples that were also analyzed for the presence of antibodies against other agents of tick-borne infections (26%, 299/1156) (Figure 1).

However, the absolute antibody titers were, on average, higher in samples tested only for *Rickettsia* antibodies compared to those being tested for additional tick-borne infection (data not shown). Most of the investigated samples (71%, 2013/2823) were collected in hospitals, primarily from departments of infectious diseases (49%; 1383/2823). Other departments included dermatology, neurology, and others.

### 3.3. Age and Gender Distribution of Patients with Rickettsia Antibodies

In the study period, 769 positive samples represent 591 patients with confirmed *Rickettsia* infection. Serology testing was the most common method used for the diagnosis of *Rickettsia* infections. Overall, the infected individuals were almost evenly distributed with regard to sex, as 51% (299/591) of the patients were male (Figure 2). The age group >30–60 included 55% (325/591) of the patients. The median age was 47 years (ranging from 1 year to 86 years). Children below the age of 18 had a lower positivity rate compared to adults.

### 3.4. Geographical Distribution of Danish Cases

Accurate determination of the geographical distribution of cases within Denmark was difficult due to the large area that each hospital covers. Only 11% (61/561) of the seropositive individuals had reports of foreign travel. Of the individuals found positive for rickettsiosis, only 5% (31/602) reported a previous tick bite.

### 3.5. Seroprevalence of Rickettsia spp. in Denmark

The serology test included IgG and IgM analysis for both SFG and TG (Table 2). Out of the 2404 serology tested samples, 726 samples were positive for either IgG and/or IgM, representing 561 individuals with antibodies against *Rickettsia*. Only 4% (81/2075) of the patients had two subsequent serology samples tested 14–31 days apart, even though follow-up is highly recommended for serology.

### 3.6. Concordance Between PCR and Serology

Overall, of the 358 individuals tested by PCR, only 193 were followed up with a serology test within 3 months of the PCR test. During this period, 119 individuals tested negative by both PCR and serology, but no follow-up sample was submitted within a 7-day period. Eight individuals were positive by PCR but were serologically negative; four of these were identified as *R. africae* infections. No follow up serology was done.

Fifty-five individuals tested positive by serology but were negative by PCR. In 14 individuals, PCR was performed from 4 days up to 4 months after the serological test. The unpaired samples for PCR and serology is a weakness for the correct diagnosis of rickettsial infections. Positive PCR tests are not followed up for antibodies and vice versa. Often, the serological testing is performed late in the course of the disease, and naturally, the PCR will be negative since the bacterium is not present.

### 3.7. Co-Infections

Out of the 2404 serology-tested samples, 1156 samples were tested for additional antibodies from relevant tick-borne infections. A total of 299 samples representing 279 individuals were positive for *Rickettsia* spp.; 10 of those individuals showed an immune response to at least two other species, indicating possible co-infections. Most co-infections were between *Rickettsia* spp. and *Bartonella* spp., or *Rickettsia* spp. and *Borrelia* spp. (Table 3).

## 4. Discussion

Scandinavian ticks are reported to be infected with *Rickettsia*, yet only a few Scandinavian human case reports of rickettsiosis are published in the scientific literature [6,11,14,21]. A report from the European Centre for Disease Prevention and Control (ECDC) concludes that rickettsiosis is poorly understood but is considered an emerging disease in Europe, and reliable data on disease risk are lacking. The report also concludes that it is not possible to prioritize rickettsiosis over other emerging diseases [16].

Accordingly, improved surveillance of rickettsiosis throughout Europe is necessary. Denmark does not have a surveillance program for rickettsiosis, and SSI is the only laboratory in Denmark that tests for human *Rickettsia* infections. Therefore, this retrospective study is nationwide and elucidates the number of Danish *Rickettsia* infections and the distribution by age and gender, together with estimates of co-infections.

In this study, we found an average seroprevalence of 26% (561/2147 seropositive samples: Figure 1) for *Rickettsia* antibodies in the study period from 2008 to 2015. In the interpretation of the serological tests, the positive cut-off value was adjusted with respect to the Danish background antibody level in a healthy population [18]. In this study, very few (approximately 4%; data not shown) of the individuals had repeated serology testing between 14 and 21 days later, which emphasizes the need to stress the importance of correct diagnosis to medical doctors. A confirmed case of acute rickettsiosis detected by serology requires seroconversion testing in a second sample, with approx. 2–3 weeks between samplings [17].

Analysis for the presence of *Rickettsia* DNA by PCR and sequencing revealed that skin samples and biopsies were superior to blood samples. Our results confirm previously described findings of low PCR sensitivity in serum samples [22], as just 6% (43/726) of the positive samples were found by PCR.

There is an emerging consensus that ticks may carry several pathogens (see [7] for a recent Danish study), but there are only a few available reports on co-infections in humans acquired from tick vectors. In one Polish study, 110 patients with tick-borne encephalitis (TBE) were analyzed for coinfections, and the authors found 30 and 12 PCR-positive patients for *Borrelia* spp. and *A. phagocytophilum*, respectively [23]. In another Polish study, a single case of *A. phagocytophilum* and a single case of *Babesia* spp. co-infections were found in 24 *Borrelia* spp.-positive samples [24]. In a Danish study, 168 patients tested positive for antibodies against *R. helvetica*; the study found that 21 patients (12.5%) also had antibodies against *R. helvetica* [11].

In the present study, indications of co-infections of *Rickettsia* with either *Borrelia* spp. (14%; 40/279) or *Bartonella* spp. (8% 22/279) were found. Others have reported that co-infections of *Babesia* spp. and *Borrelia* spp. resulted in prolonged and more intense clinical manifestations than those resulting from infection with *Borrelia* alone [25]. It is possible that the variety of clinical symptoms and outcomes in patients highly depends on the presence of co-infections.

Our data show that hospitals have submitted more samples for *Rickettsia* testing than general practitioners. This indicates a greater degree of rickettsiosis awareness among hospital personnel or that the patients arriving at the hospital have severe and clear symptoms indicative of a need for testing. It could also reflect the possibility that patients visiting general practitioners present with fewer/milder symptoms, or they are in the early stages of the disease.

Tick-borne rickettsiosis (due to the wide range of symptoms) can be mistaken for medical conditions such as viral gastroenteritis, upper respiratory tract infection, non-*Rickettsia* bacterial sepsis, idiopathic vasculitis, and viral or bacterial meningoencephalitis [26,27]. Despite non-specific initial symptoms of tick-borne rickettsiosis (e.g., fever, malaise, and headache), early consideration in the differential diagnosis and empiric treatment is critical to preventing poor outcomes, especially for Rocky Mountain spotted fever, which progresses rapidly without treatment [3]. The dermatologic classification of the rash, its distribution, a pattern of progression and timing relative to the onset of fever, and other systemic signs provide clues to help guide the differential diagnosis [17]. Recent reports from Asia and the Mediterranean region show that *Rickettsia* species (such as *R. acarii* and *R felis*) can also be transmitted by mites, fleas, and mosquitos [28,29,30]. Description of the rash after a tick bite or other insect bite, supported by the timely collection of samples for PCR and serology, will ensure the correct diagnosis of vector-borne infections. A limitation of the study is that sparse information regarding both the clinical condition and travel exposure was given along with the patient information. Clinical information could allow for correlation with diagnostic tests and calculation of known risk factors. Information on travel exposure would also make it possible to calculate a correct estimate of *Rickettsia* infections acquired in Denmark by excluding all travel-related infections. It is, therefore, important to be informed about travel and the clinical picture of the individual [3].

The present retrospective study from Denmark on laboratory evidence for SFG infections underlines the importance of alertness and diagnostic preparedness for adequate handling of rickettsial infections.

## 5. Conclusions

In conclusion, seropositivity of 26% for *Rickettsia* antibodies was observed in Danish individuals suspected of tick-borne disease. This high prevalence suggests that *Rickettsia* is endemic in Denmark, and, as previously described, patients are presenting with mild symptoms. The test can be used in a differential diagnosis for other vector-borne infections. Improved utilization of appropriate diagnostic tests, documentation of epidemiological factors, and timely reporting to public health officials will guide prevention messaging and shape public health policies.

We will stress the need for increased public awareness and strengthening of *Rickettsia* surveillance, as *Rickettsia* infections could result in an increasing disease burden as people travel more, and concerns about an increasing number of immunocompromised individuals are rising. Therefore, we recommend that both clinical data and specific travel exposure be included in the surveillance system of *Rickettsia* infections.

## Figures and Tables

**Figure 1 biology-09-00133-f001:**
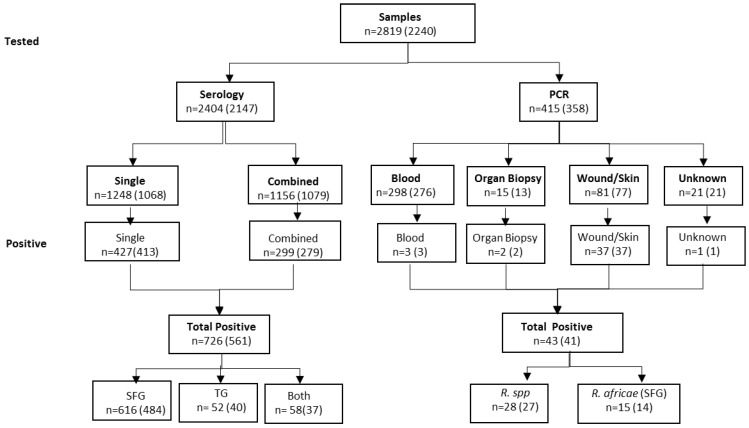
The number of tested samples and the number of positive samples detected by immunofluorescent assay (IFA) and PCR from 2008 to 2015. n: tested samples (number of patients).

**Figure 2 biology-09-00133-f002:**
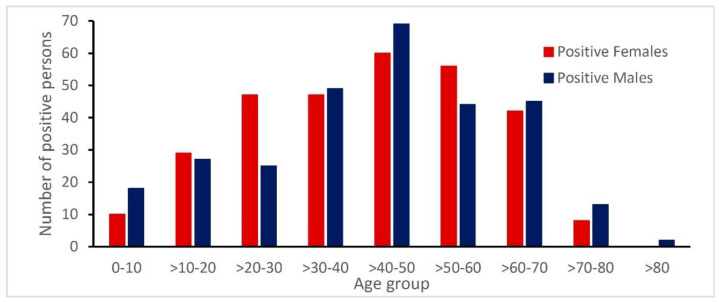
Age and gender distribution of confirmed *Rickettsia*-infected patients.

**Table 1 biology-09-00133-t001:** Distribution of samples submitted to Statens Serum Institut and tested for rickettsiosis.

		Total No of Samples		General Practitioners		Medical Specialists		Hospitals		Others ^2)^	
		Tested	n (%) ^3)^	Tested	n (%)	Tested	n (%)	Tested	n (%)	Tested	n (%)
by analysis	Single **^1)^**	1248	427 (34)	157	53 (34)	15	2 (13)	1068	368 (34)	1248	2 (19)
	Combined	1156	299 (26)	459	129 (28)	74	17 (23)	607	150 (25)	16	2 (19)
	PCR	415	43 (10)	63	13 (21)	7	0	338	30 (9)	7	0
by year	2008	29	11 (38)	9	2 (22)	2	2 (100)	18	7 (39)	0	0
	2009	469	114 (24)	196	53 (27)	20	4 (20)	250	55 (22)	3	2 (67)
	2010	411	91 (22)	132	30 (23)	7	1 (4)	268	59 (22)	4	1 (25)
	2011	404	101 (25)	85	24 (28)	10	0	302	77 (25)	7	0
	2012	354	125 (35)	67	31 (46)	3	1 (33)	280	91 (33)	4	2 (50)
	2013	332	142 (43)	58	23 (40)	6	4 (67)	265	115 (43)	3	0
	2014	418	98 (23)	74	19 (26)	16	5 (31)	325	74 (23)	3	0
	2015	402	87 (22)	58	13 (22)	32	2 (6)	305	70 (23)	7	2 (29)
Total		2819	770 (27)	679	195 (29)	96	19 (20)	2013	548 (27)	31	7 (23)

^1)^ Distribution of submitted samples for a requested analysis (Single: serological test for *Rickettsia* spp.; Combined: serological test for *Rickettsia* spp. and additional species: *Bartonella*, *Anaplasma*, *Francisella*, and *Borrelia* spp.; PCR: real-time PCR assay specific for *Rickettsia* spp; ^2)^ Ear specialist, region or other public or private customers, and unknown; ^3)^ n: number of positive samples (% of positive samples of tested samples).

**Table 2 biology-09-00133-t002:** Distribution of the number of 561 individuals according to serology detection.

	SFG		TG	
	n (%) ^1)^		n (%)	
Positive	484	(86)	40	(7)
IgM	401	(83)	18	(46)
IgM and IgG	44	(9)	1	
IgG	39	(8)	21	(54)

^1)^ n: Number of positive individuals (% of positive). Thirty-seven (7%) individuals had antibodies to both SFG and TG (inconclusive).

**Table 3 biology-09-00133-t003:** Co-infections with *Rickettsia spp*. and other tick-related infections **^1).^**

No. of Co-Infections	*Rickettsia* (R)	*B. henselae* *B. quintana* ^2)^	*B. burgdorferi* ^2)^	*Ehrlichia* ^3)^	*F. tularensis* ^3)^
0	197				
1	72	17	32	15	8
2	4	4	4		
2	3		3	3	
2	1	1		1	
2	1			1	1
2	1		1		1
Total	279	22	40	20	10

^1)^ Two-hundred-seventy-nine individuals who tested positive for *Rickettsia* spp. (IgG and/or IgM) were combination tested for *Bartonella*, *Ehrlichia*, *F. tulearensis*, and *B. burgdorferi* antibodies. ^2)^ Tested with IgG and/or IgM. ^3)^ Tested with IgG only.

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
