# Peer review of "Laboratory Diagnostics of Rickettsia Infections in Denmark 2008–2015"

_biology, 2020, doi:10.3390/biology9060133_

Round 1

Reviewer 1 Report

This is a good paper.

However.

I suggest that some more space is given in the introduction as to the clinical picture of Rickettsia infections. The next last paragraph in the discussion can be rebuilt, and moved to the introduction.

Table 1 is hard to interpret. It needs more explanation about what the figures represents.

Present percentages with numbers in parenthesis all over in the text, e.g: 32% (13/41).

In the introduction:

Rebuild/uncelar:

We found that in the Danish individuals tested positive for Rickettsia by serology,….

…giving an estimated 27% (se above) annual prevalence among those suspected of rickettsiosis after tick bites.

Author Response

This is a good paper.

However.

I suggest that some more space is given in the introduction as to the clinical picture of Rickettsia infections.

# We have added “Rickettsia spp belonging to the SFG …” starting @ line 45. We have further sharpened a few more sentences in the introduction. Notice line numbers refer to the version with “Track changes” function activated.

The next last paragraph in the discussion can be rebuilt, and moved to the introduction.

# We have considered this suggestion. We find that this paragraph also adds to the importance SFG surveillance.
We have strengthened the introduction with the severity of various Rickettsia spp caused diseases (lines 45 -51) and the importance of surveillance @ lines 77 – 84.
We have added additionally the information on R.acarii and R felis.@ lines 255 – 259 also to strengthen that importance.

Table 1 is hard to interpret. It needs more explanation about what the figures represents.

# we have added clarifying Table notes.

Present percentages with numbers in parenthesis all over in the text, e.g: 32% (13/41).

# done throughout the manuscript. A rework with the percentage numbers revealed a few calculation errors: correcting 27 % to 26 % (lines 29; referring to lines 219 and 272, that were correct in the first submission) and two obvious calculation errors in line 239.
We thank the reviewer for pointing this out.

In the introduction:

Rebuild/uncelar:

We found that in the Danish individuals tested positive for Rickettsia by serology,….
…giving an estimated 27% (se above) annual prevalence among those suspected of rickettsiosis after tick bites.

# we have sharpened the sentence @ line 24:

“Among the PCR positive individuals, 32% (13/41) were confirmed by sequencing and all of these were identified as R. africae, indicating infections after travel exposure.”

To:

“In contrast we could confirm 13 of 41 (32%) PCR positive individuals by sequencing and identify all of these as R. africae, indicating infections after travel exposure. “

In that paragraph of the abstract to clarify.

# We thank this reviewer for this constructive critique.

Reviewer 2 Report

The paper entitled: „Laboratory diagnostics of Rickettsia infections in 3 Denmark 2008–2015” is very well-written and aims at important topic of Rickettsia infections with the emphasis of underestimation in Denmark. I have only few remarks:

- Line 65 – please add, that it is underestimated not only in trvallers coming back from Africa, but also south Europe, where incidence is much higher than in the north of globe (of course there are other species than R. africae)

- Line 75, 83, 94 – please add a number of patients included to the study

- Do Authors have data regarding clinical pictures of patients with positive samples? It is written in Materials and methods: „clinical information (when provided)”, but I can not see analysis in the Results

- Please discuss why there was a big disreptancy between serological and molecular tests results?

Author Response

The paper entitled: „Laboratory diagnostics of Rickettsia infections in 3 Denmark 2008–2015” is very well-written and aims at important topic of Rickettsia infections with the emphasis of underestimation in Denmark. I have only few remarks:

- Line 65 – please add, that it is underestimated not only in trvallers coming back from Africa, but also south Europe, where incidence is much higher than in the north of globe (of course there are other species than R. africae)

# this information has been included @ lines 77-78. The complete paragraph has been sharpened (lines 77-84)
Notice line numbers refer to the version with “Track changes” function activated.

- Line 75, 83, 94 – please add a number of patients included to the study

# the sentence “In total 2819 samples obtained from 2240 individuals during this period were analyzed (Fig 1).”  was added to the end of section 2.1. (lines 96-97)

- Do Authors have data regarding clinical pictures of patients with positive samples? It is written in Materials and methods: „clinical information (when provided)”, but I can not see analysis in the Results

# we have added “Clinical and travel data were analyzed and presented by Ocias et al” @ line 93; unfortunately we have very little additional clinical information.

- Please discuss why there was a big disreptancy between serological and molecular tests results?

# we have added “The unpaired samples for PCR and serology is a weakness for the correct diagnosis of rickettial infections. Positive PCR tests are not followed up for antibodies and vice versa. Often the serological testing is performed late in the course of disease and naturally the PCR will be negative since the bacterium is not present.” To the end of section 3.6 (lines 193 - 196) in order to discuss this discrepancy.

# We thank this reviewer for this constructive critique.

Reviewer 3 Report

Rickettsioses are emerging disease globally. Epidemiological facts are scarce. The present retrospective study from Denmark on laboratory evidence for SFG infections, underlines the importance of alertness and diagnostic preparedness for adequate handling of rickettsial infections. Thus, also the possibility of other vectors, mites (R.acarii), fleas and possibly mosguitos (R.felis) could be mentioned.

In the methodology, mentioning the performance parameters of used gltA PCR and the IgG IFA at cut-off 1/512 would add to understanding of the results.

Line 127 is unclear "...tested both by serology and PCR --positive with both serology"

"Co-infections" in this context refer to positive serology, which is not evidence of an ongoing infection. Therefore, the sentence "ten of those individuals were co-infected with at least two other species" should be reformulated. 

Author Response

Rickettsioses are emerging disease globally. Epidemiological facts are scarce. The present retrospective study from Denmark on laboratory evidence for SFG infections, underlines the importance of alertness and diagnostic preparedness for adequate handling of rickettsial infections. Thus, also the possibility of other vectors, mites (R.acarii), fleas and possibly mosguitos (R.felis) could be mentioned.

# we have added “Recent reports from Asia and the Mediterranean region show that Rickettsia species (such as R. acarii and R  felis), can also be transmitted by mites, fleas and mosquitos [28-30]. Description of the rash after a tick bite and or other insect bite supported by timely collected samples for PCR and serology will ensure the correct diagnosis of vector borne infections.” Near the end of the Discussion (lines 255 - 259) to emphasize the importance of alertness to new rickettsial infections.
Notice line numbers refer to the version with “Track changes” function activated.

In the methodology, mentioning the performance parameters of used gltA PCR and the IgG IFA at cut-off 1/512 would add to understanding of the results.

# we have added “according to Stenos et al.” (now [20]) (line 111) to clarify that we are using the same parameters and the IgG IFA cutoff’s are discussed in ref Kantsø et al. 2009 (now reference [18]), as mentioned already (line 102).

Line 127 is unclear "...tested both by serology and PCR --positive with both serology"

# we agree; this sentence has been completely removed (line 143)

"Co-infections" in this context refer to positive serology, which is not evidence of an ongoing infection. Therefore, the sentence "ten of those individuals were co-infected with at least two other species" should be reformulated. 

# we agree: we have changed the sentence to:
”ten of those individuals showed immune response to at least two other species indicating possible co-infections.” (lines 200 - 201)
and in the Discussion we have emphasized the reason for investigating this by adding (lines 230 -231):
There is an emerging consensus that ticks may carry several pathogens, see [7] for a recent Danish study, but”
And emphasized that presence of immune response to several antigens is only an indication for co-infections (line 238):
“In the present study, co-infection” to “In the present study, indication for co-infection”

# We thank this reviewer for this constructive critique.